# Description-Based Text Similarity

**Shauli Ravfogel**[1]  **Valentina Pyatkin**[2,3] **Amir DN Cohen**[1]  **Avshalom Manevich**[1]  **Yoav Goldberg**[1,2]
[1]Bar-Ilan University  [2]Allen Institute for Artificial Intelligence  [3]University of Washington
{shauli.ravfogel, valpyatkin, amirdnc, avshalomman, yoav.goldberg}@gmail.com

## Abstract

Identifying texts with a given semantics is central for many information seeking scenarios. Similarity search over vector embeddings appear to be central to this ability, yet the similarity reflected in current text embeddings is corpus-driven, and is inconsistent and sub-optimal for many use cases. What, then, is a good notion of similarity for effective retrieval of text?

We identify the need to search for texts based on abstract descriptions of their content, and the corresponding notion of *description based similarity*. We demonstrate the inadequacy of current text embeddings and propose an alternative model that significantly improves when used in standard nearest neighbor search. The model is trained using positive and negative pairs sourced through prompting a LLM, demonstrating how data from LLMs can be used for creating new capabilities not immediately possible using the original model.

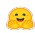 https://github.com/shauli-ravfogel/descriptions

🤗 https://huggingface.co/datasets/biu-nlp/abstract-sim

## 1 Introduction

Searching for texts based on their semantics is important for knowledge seeking agents. Such agents can be human users, or artificial ones: either LLM-based agents that are tasked with a complex goal and need to locate information as a sub-goal, or as components in retrieval augmented generation (Khandelwal et al., 2019; Guu et al., 2020; Parisi et al., 2022). Current semantic search solutions are based on dense encoders (Reimers & Gurevych, 2019; Gao et al., 2021) which learn a representation space such that "similar" documents are proximate in space. The notion of similarity in this context, however, is not explicitly defined but rather learned from vast datasets containing pairs of texts labeled as similar, often *mixing* various different kinds of similarity (Kaster et al., 2021; Opitz & Frank, 2022). This makes them sub-optimal for information seeking queries, as it is hard to control or predict the results of a given similarity-based query. What is a good query representation and similarity definition for a semantic-search use case?

In this paper, we suggest a consistent and well-defined relation between texts, which we believe to be a useful one to encode as a vector-similarity metric: the relation between abstract descriptions of sentences, and their instantiations. While LLMs can identify and operate on this relation, we find that the representation spaces that emerge using common text-encoding techniques are sub-optimal for encoding it as a similarity metric. We show how to construct better embeddings for this purpose. Using LLMs, we create a dataset that captures this specific notion for similarity, and use it to train an encoder whose representation space suppresses state-of-the-art text encoders trained on orders of magnitude more data.

Our focus is in a common kind of information need which is mostly unachievable with current search techniques: retrieving texts based on a description of the content of the text. For example, in the domain of medical research, an agent might want to find sentences discussing the efficacy of a specific drug in treating a particular condition, such as "the effectiveness of drug X in managing hypertension". Or they can go more abstract, and look for "substance abuse in animals" or "a transfer of a disease between two species". Outside

of the hard sciences, one may want to search the corpus for sentences related to a historical event, such as "an important battle fought during World War II" or "a significant scientific discovery in the field of physics". In international relations research context, the agent may want to scour a corpus for "one country threatening the economy of another country", in a trading context an agent may search for "a transaction involving precious metals", and a pop-culture journalism an agent may search twitter for "a fight between two celebrities".

In all these cases, the agent is not interested in a definition or a single answer, but in sentences whose content is a specific instantiation of their query (for example, "The studies have shown that a sub-population of primates chronically consume intoxicating amounts of alcohol" for the "substance abuse in animals" query). In other words, we are interested in a higher-order similarity reflecting the "instance-of" property.

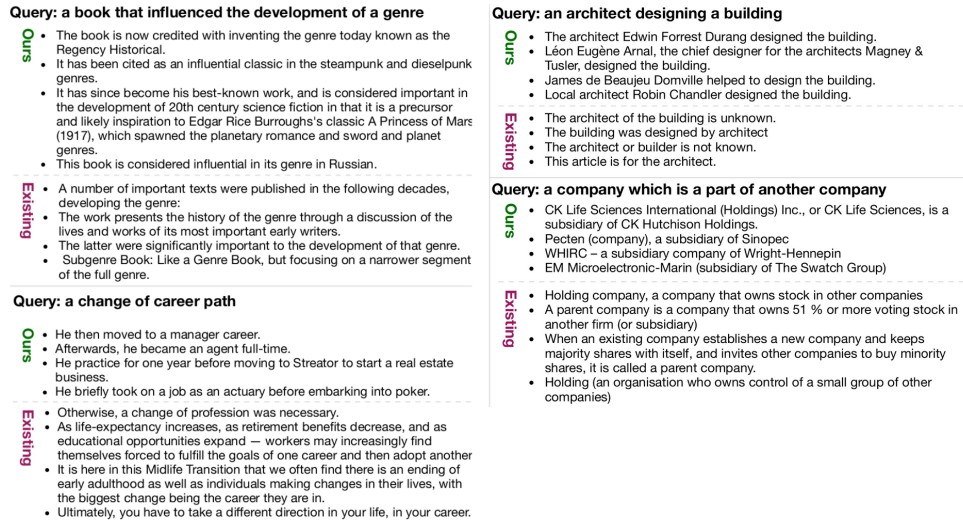

Figure 1: Top retrieval results from the Wikipedia Index. **Ours**: the model developed in this work. **Existing**: `all-mpnet-base-v2`, a strong sentence-similarity encoder.

Such retrieval cannot be easily achieved through keyword-based retrieval, because the retrieved text is more specific than the description, causing very low lexical overlap. It is also not easily achievable by current "dense retrieval" systems that rely on vector similarity: generic sentence similarity methods (Reimers & Gurevych, 2019; Gao et al., 2021) tend to retrieve texts that are similar to the description, rather than instantiations of it (e.g., a query like "*an architect designing a building*" should return a sentence like "*The Fallingwater, a remarkable architectural masterpiece located in rural southwestern Pennsylvania, was designed by Frank Lloyd Wright*"" and not "*The architect participates in developing the requirements the client wants in the building.*", although the latter is more similar under conventional sentence similarity models). Similarly, systems that are trained to retrieve passages that contain answers to questions (trained, for example, on SQuAD (Rajpurkar et al., 2016; 2018)), beyond being focused on questions rather than assertions, are also focused on specifics rather than abstract situations (questions are of the form "*when did Columbus discover America*" and not "*a discovery on a new land by an explorer*"). Models trained on large data from search query logs may be more diverse, but are generally not available outside of a few large technology companies. We do not find an existing soltuion that fits our goal (Section 6).

We show that retrieval based on description is achievable: given training data consisting of `<description, text>` pairs, we can train a descriptions encoder and a text encoder that learn to represent items such that the descriptions and the texts they describe are close in embedding space (§4). These vector encodings can then be used in a standard similarity-based retrieval setting. Figure 1 shows four queries that did not appear in the training data, and their top-4 retrieved items, over a corpus of almost 10M wikipedia sentences.

To obtain the training data (§3), we observe that the reverse direction of the process, going from a text to its description, is a task that can quite easily be performed either by crowd-workers, or, as we do in this work, by large language models such as GPT-3 (Brown et al., 2020) and Codex (Chen et al., 2021). We thus use the davinci-text-03 model to generate descriptions of sentences sampled from Wikipedia, and use the result as our training corpus. Each sentence can accommodate many different descriptions, pertaining to different aspects of the text. We therefore produce five different descriptions for each text, in addition to incorrect descriptions, to be used as negative examples. We find the models trained on this data to excel in both human (Section 5.1) and automatic (Section 5.2) evaluation.

## 2 Description-based Similarity

General similarity metrics in document retrieval capture a broad range of lexical, syntactic, and semantic resemblances and relations, offering a foundational approach to similarity assessment across various tasks. However, their overarching nature often compromises task-specific relevance and precision. In contrast, we propose a specific relation that we want to be reflected by the similarity metric–*the abstract description relation*–which explicitly models the relation between high-level descriptive queries and concrete instances within documents.

We start by defining the abstract description relation:

**Definition 1 (The Abstract-Description Relation)** *Given two texts,*[1] *T and D, we say that* $(T, D)$ *satisfies the* abstract description *relation iff:*
*1) D describes*[2] *(some of) the content of T.*
*2) D contains less information and is less specific than T.*

Note that this is a many-to-many relation, which is not reflexive, anti-symmetric and non-transitive. The *abstract descriptions* relation relates to, but is not the same, as other text-based semantic notions such as *paraphrases*, *entailments* and *summaries*. In particular, descriptions are not paraphrases, as paraphrases are symmetric and non lossy. The description relation is also more specific than summaries or entailments: while many descriptions are also participating in the entailment and summary relations, not all entailments or summaries are abstract descriptions. One notable difference from summaries is that summaries attempt to capture the main events of the text, and do not abstract over it.

**Example** To illustrate the abstract-description relation, consider the following text and the three valid descriptions of it (taken from our dataset, Section 3):

- **Text**: "On July 2, concurrent with the Battle of Gettysburg in neighboring Adams County, Captain Ulric Dahlgren's Federal cavalry patrol galloped into Greencastle's town square, where they surprised and captured several Confederate cavalrymen carrying vital correspondence from Richmond."
- **Description 1**: Military personnel thwarting an enemy's attempt to convey vital documents.
- **Description 2**: The disruption of a communication exchange in a rural area.
- **Description 3**: A dramatic, unexpected event occurring in a town square during a battle.

Clearly, the descriptions are highly abstract, in contrast to conventional summary of the text; and they omit some key details, such as the country and exact conflict being discussed, or even the fact the event occurred during a battle (description 2); the date; and the specific units being involved. Additionally, the sentence "*The town of Greencastle existed during the battle of Gettyburg*" is entailed by the text, but does not describe it. See Section 6 for additional discussion of relation to previous work.

---

[1]In the current paper, both *T* and *D* are sentences, but this not part of the definition: *T* can be londer or shorter than a sentence.

[2]We do not formally define *describes* and build on the intuitive, English language meaning of the term.

**Utility**  We argue that abstract descriptions provide a natural and efficient way to express information seeking needs: users can always *describe* the results they want in natural language. Importantly, these descriptions only need to cover *some* of the content, not every aspect (as some may be *irrelevant* to the user). A military historian might want to find dramatic events during a battle (description 3) without specifying the time or location.

**Similarity**  Existing text encoders struggle with this relation because descriptions have little lexical overlap with the text and they all *lack concrete details* mentioned in the text. However, if we could create a representation space where the text is close to each description, retrieval would be straightforward. Our goal is to learn embedding functions $E_T$ and $E_D$ for texts and descriptions such that $sim(E_T(T), E_D(D))$ correlates with the abstract desciprion relation by being higher for $T$ and $D$ for which the relation holds than for all other pairs.

## 3   Obtaining Training Data

We use GPT-3 (text-davinci-003) to generate positive and misleading descriptions for sentences from the English Wikipedia dataset.[3] For each sentence, we generate 5 valid descriptions and 5 misleading descriptions. In total, we generate descriptions for 165,960 Wikipedia sentences. See the Appendix for the exact prompts we use.

**Generating more abstract descriptions**  While the descriptions we generate do tend to be abstract, to augment the dataset with descriptions of higher abstraction, we randomly select a subset of instances, re-prompt GPT3 with three of the valid descriptions it generated, and ask it to generate abstract versions of them (this prompt is an in-context learning one, the exact prompt appears in Appendix A.1). This results in 69,891 additional descriptions for 23,297 sentences (14.3% of the data). To illustrate the effect of this iterative generation, for the sentence "Civil war resumed, this time between revolutionary armies that had fought in a united cause to oust Huerta in 1913–14.", one of the original descriptions generated was "A conflict between opposing groups arising from the overthrowing of a political leader", while the iterative query resulted in the more abstract description "Conflict arose between two sides that had previously been allied.".

**Final dataset**  Table 1 shows several examples of the generated data, including the original sentence and pairs of valid and misleading descriptions. The generated data includes a wide range of both positive and misleading descriptions that align with the original sentence and the abstract description. The positive descriptions accurately capture the main meaning and key concepts of the sentence, while the misleading descriptions contain inaccuracies or irrelevant information. We have randomly divided the data into 158,000 train, 5000 development and 2960 test instances, each composed of a sentence, 5 invalid descriptions and 5-8 valid descriptions. We found the quality of the generated descriptions adequate for training, and for measuring progress during iterative development, which we also confirmed through a human evaluation. We showed 229 valid descriptions and corresponding sentences to Turkers, asking them to rate on a scale of 4, how well the sentence fits the description. On average the instances were highly rated with a score of 3.69/4, which lies between *The sentence covers most of the main points mentioned in the description* and *The sentence covers everything mentioned in the description*.

However, some of the descriptions do not adequately capture our intended spirit of abstract descriptions of sentences that reflect an information need. Thus, for the purpose of human-evaluation of quality (Section 5), we manually curate a subset of 201 sentence descriptions from the test set, which we manually verified to reflect a clear information need that would make sense to a human. These were collected by consulting only the descriptions, without the associated sentences they were derived from, or any model prediction based on them.

---

[3]https://huggingface.co/datasets/wikipedia

| Sentence | Good Descriptions | Bad Descriptions |
|---|---|---|
| Intercepted by Union gunboats, over 300 of his men succeeded in crossing. | A large group of people overcoming a challenge. | A group of people being intercepted while crossing a desert. |
| Dopamine constitutes about 80% of the catecholamine content in the brain. | A neurotransmitter found in the brain in high concentrations. | A neurotransmitter found in the stomach in high concentrations. |
| In December 2021, Kammeraad was named in Philippines 23-man squad for the 2020 AFF Championship held in Singapore. | A sportsperson's inclusion in a squad for a championship. | A soccer player selected for a tournament in the Philippines in 2021. |
| Around this time, MTV introduced a static and single color digital on-screen graphic to be shown during all of its programming. | A visual element was implemented to enhance the viewing experience. | MTV's use of a dynamic graphic. |
| At the signing, he is quoted as having replied to a comment by John Hancock that they must all hang together: "Yes, we must, indeed, all hang together, or most assuredly we shall all hang separately". | A historical event where a significant figure made a comment about unity. | A joke about the consequences of not working together. |
| It was said that Democritus's father was from a noble family and so wealthy that he received Xerxes on his march through Abdera. | A description of a wealthy family's involvement in a significant event. | A description of a famous leader's family background. |
| Heseltine favoured privatisation of state owned industries, a novel idea in 1979 as the Conservatives were initially only proposing to denationalise the industries nationalised by Labour in the 1970s | A political party's plan to reverse a previous government's policy. | The effects of privatisation on the economy. |

Table 1: Examples of generated data training data, including the original sentence, the good and bad descriptions

## 4 Encoder Training

In order to train our model for the task of aligning sentences with their descriptions, we utilize a pretrained sentence embedding model and fine-tune it with contrastive learning. During the training process, we represent each sentence and its corresponding valid descriptions using two distinct instances of the model: one as a sentence encoder and the other as a description encoder.

Let $S$ represent a set of sentences, $P_s$ represent the set of valid descriptions associated with a sentence $s$, and $N_s$ represent the set of negative descriptions for that same sentence $s$. We encode each sentence and description via a model, resulting in a vector representation for each token. We use mean pooling over the token vectors of each of the sentence and description pairs to obtain vector representations in $\mathbb{R}^d$. Specifically, we denote the vector representation of a sentence $s$ as $\mathbf{v}_s$, the vector representation of a valid description of it as $\mathbf{v}_p$, and the vector representation of a negative description as $\mathbf{v}_n$.

To train the encoder, we combine two loss functions: the triplet loss (Chechik et al., 2010) and the InfoNCE loss (van den Oord et al., 2018) .

The triplet loss, denoted as $\mathcal{L}_{\text{triplet}}(s)$, is calculated for each sentence $s$ as follows:

$$\sum_{(p,n) \sim P_s \times N_s} \max(0, m + \|\mathbf{v}_s - \mathbf{v}_p\|^2 - \|\mathbf{v}_s - \mathbf{v}_n\|^2) \tag{1}$$

Here, $m$ represents the margin parameter that defines the minimum distance between the positive and negative descriptions. We take $m = 1$. This loss encourages the representation of each sentence to be closer to its valid descriptions than to its invalid descriptions.

The InfoNCE loss, denoted as $\mathcal{L}_{\text{InfoNCE}}(s)$, is computed using a random collection of in-batch negatives (i.e., valid descriptions of *other* sentences in the batch, as well as sentences that correspond to those descriptions). Let $N_s'$ represent the set of all in-batch negatives sampled from the valid descriptions of other sentences within the batch, including the

sentences themselves. The InfoNCE loss is given by:

$$-\log\left(\frac{\exp(\frac{\mathbf{v}_s \cdot \mathbf{v}_p}{\tau})}{\exp(\frac{\mathbf{v}_s \cdot \mathbf{v}_p}{\tau}) + \sum_{n' \in N'_s} \exp(\frac{\mathbf{v}_s \cdot \mathbf{v}_{n'}}{\tau})}\right) \tag{2}$$

Where $\cdot$ is cosine similarity and $\tau$ is the temperature (we take $\tau = 0.1$).

The final loss used for training is a combination of the triplet loss and a scaled version of the InfoNCE loss:

$$\text{Loss}(s) = \mathcal{L}_{\text{triplet}}(s) + \alpha \mathcal{L}_{\text{InfoNCE}}(s) \tag{3}$$

We take $\alpha = 0.1$. An ablation study revealed a modest improvement when using the combined loss compared to using only the triplet component or only the Info-NCE component (Appendix A.5). We train for 30 epochs with a batch size of 128 and optimize using Adam (Kingma & Ba, 2015).

## 5 Evaluation

Traditional information retrieval (IR) benchmarks do not align with our focus on abstract semantic similarity, matching generalized descriptions with explicit, concrete instances. As such, we construct a set of test queries, and quantitatively evaluate our model in two ways. We perform human evaluation on the results retrieved from a large corpus (Section 5.1). Additionally, we perform automatic evaluation on an adversarially-constructed set of relevant and irrelevant sentences for the test queries (Section 5.2), to test the robustness of our model. We attach the training and test sets, alongside the code, in the supplementary material.

**Setting** We sample a set of 10 million Wikipedia sentences (in addition to the set used for training and evaluation). We filter sentences shorter than 6 words, leaving a set of 9.55 million sentences. We encode them using the trained sentence encoder, resulting in an index called *the Wikipedia Index* henceforth. This is the set from which we retrieve in evaluation. Given a query $q$, we represent it with the query encoder and perform exact nearest-neighbor search under cosine distance.

**Evaluation set** We chose a random set of 201 descriptions from the test set, which we manually verified to be reasonable description-queries a person may be interested in. We then performed crowd-sourced evaluation of retrieval based on these descriptions, comparing our abstract-similarity model to each of the baseline models.

**Baselines** We evaluate our model against strong sentence encoder models based on the MTEB (Muennighoff et al., 2022) leaderboard in the Sentence-Transformer framework (Reimers & Gurevych, 2020),[4] all-mpnet-base-v2, E5-base (Wang et al., 2022), Instructor (Su et al., 2022), GTE-large Li et al. (2023), EmBER-v1 [5], BGE-en Xiao et al. (2023), and contriever Izacard et al. (2021) [6]. All models were finetuned by their creators on diverse sentence-similarity datasets, containing *orders of magnitude* more data than ours. Beyond the Sentence-Transformer models, our study incorporates 3 additional baselines: BM25, HyDE and a SNLI-based model (Bowman et al., 2015b). BM25 (Robertson et al., 1995) uses term frequency and document length to estimate a document's relevance to a specific query. BM25 has been shown to be a strong baseline for document retrieval (Izacard et al., 2022). HyDE (Gao et al., 2022) is a zero-shot model using GPT-3 to generate synthetic documents for a given query. The dense representations of these documents are averaged and fed as a query to a pretrained document retriever.[7] Due to the similarity between descriptions and

---

[4]https://huggingface.co/sentence-transformers

[5]https://huggingface.co/llmrails/ember-v1

[6]GTE, EmBER and BGE are the state-of-the-art in the time of the writing as per the benchmark, while the other models are highly popular.

[7]Note that HyDE is different than our model and the other baselines in the sense that it calls the GPT-3 API once per query at inference time.

entailments, we also finetune a MPnet-based model for retrieval on the SNLI dataset. See Appendix A.2 for details on our baselines.

**Our model**, denoted as `Abstract-sim`, is a fine-tuned version of the pretrained `MPnet` model (Song et al., 2020). We do not use `all-mpnet-base-v2`, which was further finetuned on similarity datasets, as it yielded worse results in preliminary experiments. Fig. 1 shows the top results of four queries, alongside the top results from `all-mpnet-base-v2` for comparison.

## 5.1 Human Evaluation

We perform human evaluation over naturally occurring sentences, in a natural retrieval scenario, where abstract descriptions are likely to be used as queries. The human evaluation compares the top sentences retrieved with our method, and to the top sentences retrieved with the state-of-the-art semantic sentence encoding models.[8]

The evaluation setup is structured as follows.[9] Crowdworkers are shown a query and results from search over the Wikipedia Index. Particularly, they are shown 10 sentences, 5 of which are the top-5 retrieved sentences from `abstract-sim` and 5 of which are the top-5 retrieved sentences from one of the baseline (each experiment with another baseline). The 10 sentences are randomly shuffled, and crowdworkers are then asked to select all sentences that they deem a reasonable fit for the query. Each task is shown to three distinct annotators. We aimed at paying crowdworkers $15 per hour on average. Each query instance is shown to 3 annotators.

**Metrics** We report the average number of results from each model that were selected as relevant (as a histogram), as well as the mean number of times a specific number of sentences from a given model was chosen (the mean of the histogram).

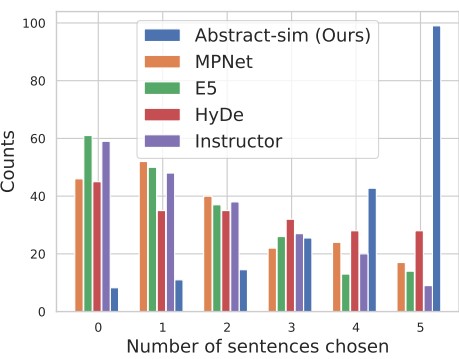

| Model | # chosen |
|---|---|
| `abstract-sim` | **3.89±0.073** / 5 |
| `HyDE` | 2.2 / 5 |
| `all-mpnet-base-v2` | 1.89 / 5 |
| `Instructor-large` | 1.64 / 5 |
| `E5-base` | 1.61 / 5 |

Table 2: Human evaluation results (Section 5.1): number of sentences that crowdworkers deemed to be fitting the query, from a set of 5 retrieved sentences: Our model (abstract-sim) vs. the four baselines. The number reported for `abstract-sim` is a mean±std over the binary comparisons against each of the 4 baselines.

Figure 2: Human evaluation results (Section 5.1): number of times a given number of sentences was chosen per query instance: Our model (abstract-sim), averaged over all 4 baseline evaluations, vs. the baselines.

**Results** For evaluation we only count sentences to have been selected as relevant, if they were chosen by at least 2 out of 3 annotators. In Table 2 we show the average number of valid retrieved sentences per method. The annotators have chosen significantly more sentences from our `abstract-sim` model compared to all 4 baselines, with our model having close to 4 out of 5 sentences deemed as fitting the query on average and the baseline models between 1.61-2.2 sentences. Fig. 2 shows the complete distribution of the number of times a given number of sentences was chosen from a given model (where the maximum is 5, that is,

---

[8]We do not compare against `NLI` and BM-25 due to their very low precision and recall in the automatic evaluation (Section 5.2; Appendix A.4) and Fig. 4. Additionally, due to budget constraints, we only compare against the top-3 dense retreieval as per the automatic evaluation: E5, MPnet and Instructor.

[9]Screenshots of the annotation interface can be found in the Appendix.

all the 5 results for the model were chosen). Notably, in 99/201 of the test cases, 5 sentences were chosen from `abstract-sim`'s results; from the baselines all 5 sentence were only chosen between 14-28 times. That is, in many of the cases all top results were considered as relevant for the query. Conversely, the baselines show a large number of cases where only 0,1, or 2 sentences where chosen, while these cases are much rarer among `abstract-sim` results (below 5 vs. at least 42 for the case of 0 relevant sentences). Overall, human inspection of top-retrieved results show a large preference for our models compared with the baselines.

## 5.2 Automatic Evaluation

We accompany the human evaluation with a manually-constructed automatic evaluation dataset, focused on robustness to misleading results. We do not know how many relevant sentences exist in the Wikipedia index for each query (if any). To allow for an automatic evaluation in the face of this challenge, we use the following evaluation scheme. We used `GPT` to generate a set of valid sentences per description. To test robustness, we work under an adversarial setting, where for each query we generate both relevant sentences and distracting sentences. We measure the precision and recall of our model and the baselines mentioned above.

**Generating sentences from descriptions** We start with the 201 valid, manually-verified descriptions in the test set. We use `GPT` for the reverse task of our main task: mapping abstract descriptions to concrete sentences. We randomly choose one negative (invalid) description from the entry in the test set that corresponds to each valid description. We manually verify that the chosen description is indeed topically similar but invalid. In case the description does not contradict the valid description, we manually change it. The process results in a complementary set of 201 invalid abstract descriptions. For example, for the valid test example "The existence of a river and a town with the same name", we have the invalid description "The existence of a river and a county with the same name". For both the valid and invalid descriptions, we generate a set of 12 sentences that match the given descriptions, ending up with 12 sentences that align with a description, and 12 sentences that align with a *contradicting* description, that serves as a distractor. These 24 sentences were then combined with the remaining 9.55 million sentences in the Wikipedia Index. The prompt used to generate these sentences is available in Appendix A.1.1. We use Mturk to verify the validity of the resulting set of sentences.[10] The process results in an average of 11.2 **valid sentences** and 9.3 **invalid sentences** per test query. See Appendix A.3 for a sample.

**Setting** We follow 3 metrics: *valid-recall@k*, *invalid-recall@k* and *precision@k*. *valid-recall@k* measures the number of valid sentences captured within the first k retrieval results over the Wikipedia index. Similarly, *invalid-recall@k* measures the number of invalid sentences captured. Finally, *precision@k* is calculated only with respect to valid and invalid sentences (excluding the Wikipedia index, which might contain many more valid sentences): we calculate the similarity of the description to the valid and invalid sentences, and count the number of valid sentences within the top $k$ results.

**Results** The precision results are shown in Appendix A.4 and the recall results are shown in Appendix A.5. Our models improves over all baselines in terms of *precision@k*. The gap is largest for *precision*@1, and gradually decreases. Our model achieves *precision*@1 = 85.4%, compared with 73.6% for the strongest baseline, E5, corresponding to 31/201 vs. 53/201 errors in the highest ranked result, respectively. The gap decreases with increasing $k$ (note that we have a maximum of 12 positive examples). As for recall, generally models that achieve high *valid-recall@k* also achieve high *invalid-recall@k*. Our model achieves relatively low *valid-recall@k*, but is better than all models in terms of *invalid-recall@k* (except SNLI, which has both low valid recall and low invalid recall), i.e., it tends to avoid returning invalid sentences, at the price of missing some valid ones.

---

[10]For the set of valid sentences we filter out all sentences chosen as fitting the description by at least two annotators, For the set of invalid sentences we take all sentences chosen as not to be a suitable fit for the description by at least two annotators.

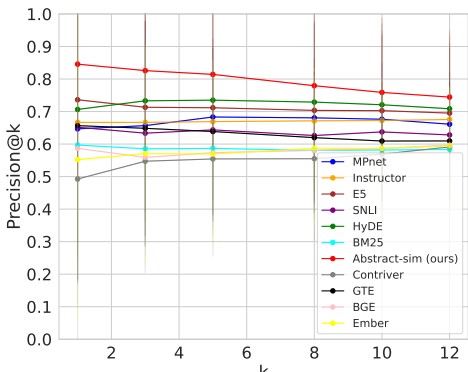

Figure 3: Precision automatic evaluation results (Section 5.2): precision@k curve for `abstract-sim` and the baselines. Vertical lines represents 1 standard deviation.

## 6    Description-based Similarity vs. Previous Work

We compare description-based similarity with popular existing similarity-based retrieval methods, as well as the related NLP task of recognizing textual entailment (Dagan et al., 2005; Bowman et al., 2015a).

**Vs. Keyword-based Retrieval:**   Keyword-based retrieval methods rely on exact lexical matches, which makes them inherently weak for retrieval based on abstract descriptions. These methods require users to construct queries using specific keywords, resulting in a laborious and suboptimal process. For example, to retrieve sentences related to "animals," a user would need to come up with an exhaustive list of animal names, which is impractical and may lead to incomplete results. Consequently, keyword-based retrieval is ill-suited for retrieving sentences based on abstract descriptions.

**Vs. Dense Similarity Retrieval:**   This family of methods, exemplified by SBERT (Reimers & Gurevych, 2019) encodes sentences based on an objective that encourages sentences with "similar meaning" to have high similarity. Similar meaning, here, is determined by multiple corpora such as Reddit comments (Henderson et al., 2019), SentEval (Conneau & Kiela, 2018) and SNLI (Bowman et al., 2015a). As such, the type of similarity captured by such models in practice emerges from the training corpus (Kaster et al., 2021; Opitz & Frank, 2022) and is not well understood. Our goal is a similarity metric for the specific type of relation we define, between abstract descriptions and concrete instantiations of them. Moreover, while sentence similarity models aim to cluster sentences with similar meaning together, a description does not have a "similar meaning" to the text it describes, but rather to other descriptions of the same text.

**Vs. QA-trained Dense Retrieval:**   These systems are trained to retrieve paragraphs based on a question, in an open-QA setting (Karpukhin et al., 2020) The retrieved paragraphs are then run through a reader component, which attempts to extract the answer from each retrieved paragraph. The training objective is to encode paragraphs to be similar to the questions to which they contain an answer. Question could be seen as similar to descriptions (e.g. "early albums of metal bands" can be served by retrieving for "which metal bands released an early album"), but they also differ in that: (a) it is often cumbersome for a user to rephrase the information need as a question—in the above example, the move to question form is not trivial; (b) questions are often focused on a single entity that is the answer to the question, rather then on a situation involving a relation or interaction between several entities; (c) the kinds of questions in current QA training sets tend to ask about specific,

rather than abstract, cases, e.g. asking "which metal band released album Painkiller?" or "what is the first album by Metallica?".

Moreover, in many cases that translation of descriptions to questions is altogether impossible. Often, there is no single question whose answer accurately fulfills the information need that can be expressed by a simple description. Consider a user interested in movie scripts where "A character is being rescued by another character". Formulating this abstract description is easy. On the other hand, while it is possible to formulate several questions that resemble that description, such as "*In what setting* is one character being rescued by another" or "*What positive help* does one character give to another character?", none of them accurately captures the intent of the original description.

**Vs Query-trained Dense Retrieval:** These systems are trained on a collection of `<query,document>` pairs, which are typically obtained from search engine logs.[11] In a sense, these subsume the description-retrieval task, but are (a) focused on documents and not on sentences; (b) not focused on this task, so may retrieve also results which are not descriptions; and, most importantly (c) are mostly based on proprietary data that is only available within a handful of large companies.

**Vs. Entailment / NLI** `<description, text>` pairs adhere to the entailment relation between positive `<hypothesis,text>` pairs in the Textual Inference task (Dagan et al., 2005; Bowman et al., 2015a), which is a superset of the `<description,text>` relation. In theory, NLI based similarity models could perform well on this task. However, in practice they do not perform well, possibly due to the composition of existing NLI datasets. Additionally, the do not usually encode the hypothesis and the premise independently, making efficient indexing difficult.

## 7 Conclusions

We introduce the task of sentence retrieval based on abstract descriptions. We show that current sentence-embedding methods are not a good fit for the task. We leverage GPT-3 to generate a set of diverse valid and invalid abstract descriptions, and train a retrieval model on that resulting data. We find that the model trained on the data that is tailored to this task is performing significantly better than standard sentence-similarity models. This disparity highlights that the notion of similarity captured by sentence transformers is vague, and that tailoring it to specific information seeking need may result in significant practical improvements.

## Acknowledgements

This project received funding from the European Research Council (ERC) under the European Union's Horizon 2020 research and innovation program, grant agreement No. 802774 (iEXTRACT). Shauli Ravfogel is grateful to be supported by the Bloomberg Data Science Ph.D Fellowship.

## Ethics Statement

As all language technology, the models and data are inherently dual use—they can be used both for good (e.g., to advance human knowledge) or for bad (e.g., for surveillance that is aimed at depression of minority communities). We hope that the benefits outweighs the risks in our case.

---

[11]In the context of academic research, the focus is on the MSMARCO dataset (Bajaj et al., 2016), which contains natural language questions extracted from query logs. However, query logs include many different query types beyond questions, and modern search systems have been reported to incorporate such embedding based results for general queries.[12]. For these, see "Vs. QA-trained dense retrieval" above.

According to the terms-of-service of the GPT API, the API output (the collected data and the models we created based on it) should not be used to compete with OpenAI. We declare we have no such intentions, and ask the users of the data and models to also refrain from doing so.

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

# A  Appendix

## Limitations

Our training data, models and experiment are all strictly English-based. More importantly, we observed the following limitation of the resulting similarity model. While it clearly is better than all existing models we compared against at identifying sentences given an abstract description, we also observed the opposite tendency: for some queries, it is not faithful to the provided description. For example, searching for the query "The debut novel of a french author" returns results such as "Eugénie Grandet is a novel first published in 1833 by French author Honoré de Balzac" or "Lanzarote (novel), a novel by Michel Houellebecq", either mentioning the first time the novel was published, instead of returning mentions of a first novel published by an author; or mentioning novels written by French authors, regardless of whether or not they are their debut novels.

## A.1  Prompting

These are the prompts we used to generate the sentence descriptions dataset. The "main prompt" was used to generate 5 valid descriptions and 5 invalid descriptions per sentence. For approximately 14% of the sentences, we re-feed GPT with one of its valid generations and use the "Make-more-abstracts prompt" to generate 3 additional more abstract version of the descriptions. Finally, we use the "Description to sentence prompt" to generate a set of sentences that align with the 201 test descriptions, used for evaluation.

**Main prompt:**

```
Let's write abstract descriptions of sentences. Example:

Sentence: Pilate 's role in the events leading to the crucifixion lent themselves
to melodrama , even tragedy , and Pilate often has a role in medieval mystery
plays .

Description: A description of a historical religious figure's involvement in a
significant event and its later portrayal in art.

Note: Descriptions can differ in the level of abstraction, granularity and the
part of the sentence they focus on. Some descriptions neeed to be abstract, while
others should be concrete and detailed.

For the following sentence, write up 5 good and stand-alone, independent
descriptions and 5 bad descriptions (which may be related, but are clearly wrong).
Output a json file with keys 'good', 'bad'.

Sentence: {sentence}

Start your answer with a curly bracket.
```

### A.1.1  Make-more-abstract Prompt

```
Sentence: in spite of excellent pediatric health care , several educational
problems could be noted in this tertiary pediatric center .

Description: Despite having advanced healthcare resources, certain deficiencies
in education were identified at a medical center that serves children.

A very abstract description: The provision of care at a specialized medical
center was not optimal in one particular area, despite the presence of advanced
resources.
```

```
Sentence: {sentence}

Description:  {description}

A very abstract description:
```

**Description to sentence prompt**

```
Create a JSON output with a key 'sentences' containing 15 Wikipedia-style different
sentences. The sentences should align with the given description, i.e., the
description must be a valid characterization of the sentences. Notice: (1) You
must avoid using words appearing in the description; (2) You MUST mention concrete
entities such as names of people, places and events to make the sentence sound
natural; (3) you MUST make sure each sentence is relevant for the description;
(4) IMPORTANT: you MUST make the sentences different from each other; they must
not mention the same topics. Description: '{description}'

Be faithful to the description. Start your answer with a curly bracket.
```

## A.2   Baseline Models

**HyDE**   We adapted HyDE to our scenario by: a. adding an appropriate prompt for sentence generation matching the description in the query and b. replacing the document retriever with a sentence retriever (`all-mpnet-base-v2`).

**Instructor**   `Instructor` generates task-specific embedidngs by specifying the type of task in the prompt. We use the recommended prompt "Represent the Wikipedia document for retrieval" for the sentence encoder, and the closest prompt from Su et al. (2022)'s dataset, "Represent the Wikipedia summary for retrieving relevant passages:", for the description encoder; variations on the query prompt, such as "Represent the Wikipedia description for retrieving relevant passages:", yield similar results.

**SNLI baseline**   The notion of description-based similarity is related to NLP task of recognizing textual entailment (Dagan et al., 2005; Bowman et al., 2015a) (see below in Section 6). As such, it is natural to ask how do models trained on popular RTE datasets, such as SNLI (Bowman et al., 2015b), fare on this task. We extract entailment and neutral pairs from the SNLI dataset, and finetune an `MPnet-base` model for 30 epochs with the objective of minimizing the InfoNCE loss Eq. (2), where hypothesis is the query, the negative pairs are taken from neutral premises while the positive is the entailing premise. We then evaluate this model in the same way we evaluate the other baselines.

## A.3 Automated evaluation data

| Description | Valid sentence | Invalid sentence |
|---|---|---|
| A period of difficulty and sorrow for an individual. | The death of his beloved mother was an extremely difficult and sorrowful time for Albert. | This individual building was a difficult place to live in. |
| A shift in the way people are referred to has occurred. | In the current era, more and more people are preferring to go by their given name, rather than traditional titles. | Alice was referred to as 'miss' the same way she used to be in the pre-quarantine period. |
| The honoring of an actor's legacy. | On 10 April 2020, a ceremony was held at the TCL Chinese Theatre in Hollywood to commemorate the late actor Peter O'Toole, who passed away in 2013. | Thespians from all over the nation had gathered in Los Angeles to recognize the immense influence of veteran director Stan Li. |
| The act of two individuals reaching a mutual understanding. | The two leaders of different nations decided to set aside their differences and reach a peaceful understanding. | Three high school friends, Alex, Jack, and Rachel, finally reached a mutual agreement over which dessert they'd order at the cafeteria. |
| A dismissal of a concept by a renowned scholar. | Although Albert Einstein highly esteemed science, he strongly denied the possibility of perpetual motion. | The acclaimed academic disassociated himself from the researcher he had once championed. |
| A federal grand jury's investigation into a political corruption case. | A federal grand jury has launched an investigation into a political corruption scandal involving prominent figures in the government. | The federal grand jury is conducting a thorough investigation into the devastating floods that occurred across the nation. |
| HeThe effect of a decrease in the number of predators. | The declining trend in the number of predators has caused a severe depletion in the prey population. | Predators have evolved over time, playing a critical role in ecology, occupying different niches and competing with each other. |

Table 3: Examples of generated training data, including the original sentence, the good and bad descriptions

Table 3 presents a sample of descriptions from the 201 examples test set, alongside one invalid and one invalid sentence (generated by GPT3) per description. These were used in the automatic evaluation (Section 5.2).

## A.4 Recall Results

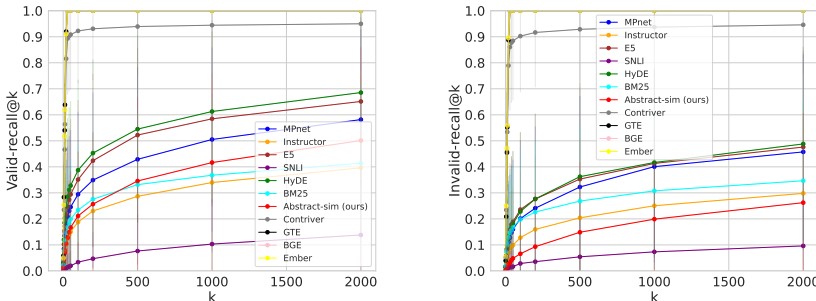

Figure 4: Recall automatic evaluation results (Section 5.2): valid-recall@k (left, higher is better) and invalid-recall@k (right, lower is better) for `abstract-sim` and the baselines. Vertical lines represent 1 standard deviation.

Figure Fig. 4 presents valid-recall@k (higher is better) and invalid-recall@k (lower is better) for the automatic evaluation experiment (Section 5.2).

## A.5 Ablation of Loss Components

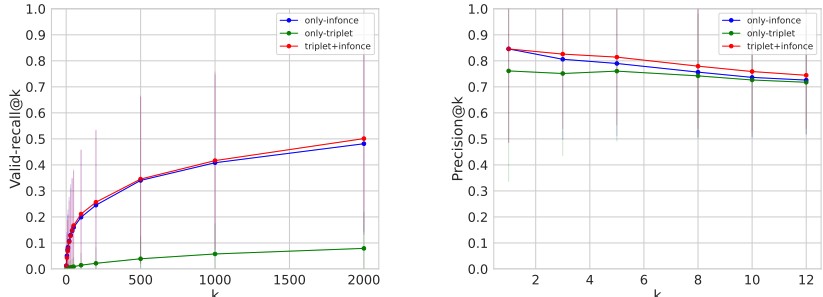

Figure 5: Ablation results on the automatic evaluation (Section 5.2).

Fig. 5 presents the results of automatic evaluation when training models with the individual loss components (only the triplet loss, or only the info-NCE loss) compared with using the combination of the two losses.

## A.6 Human-evaluation Interface

This is the interface used for MTurk evaluation:

---

**Instructions (click to expand/collapse)**

Thanks for participating in this HIT! Please read the instructions carefully.

In this HIT, you will be shown a **Description** and 10 **Sentences**. The **Description** details what type of sentence we are looking for: Imagine the description is something like a search query for a search machine. The **Sentences** is the result we obtained after searching for sentences that fit the description.
Your task is to *choose* all **Sentences** which you consider good matches for the Search Query/**Description**. Note that the **Sentence** is allowed to contain additional information, not mentioned in the **Description**, as long as it covers what has been requested in the **Description**.

Please take care to not submit responses that are uninformed by the instructions.

---

**Description:**
${description1}

**1.** Choose all **retrieved sentences** that fit the **description**

☐ ${sentence1}
☐ ${sentence2}
☐ ${sentence3}
☐ ${sentence4}
☐ ${sentence5}
☐ ${sentence6}
☐ ${sentence7}
☐ ${sentence8}
☐ ${sentence9}
☐ ${sentence10}

(Optional) Please let us know if anything was unclear, if you experienced any issues, or if you have any other fedback for us.

---

This is the interface with an instantiated descriptions and 10 retrieved sentences (5 from baselines and 5 from our model, presented in random order).

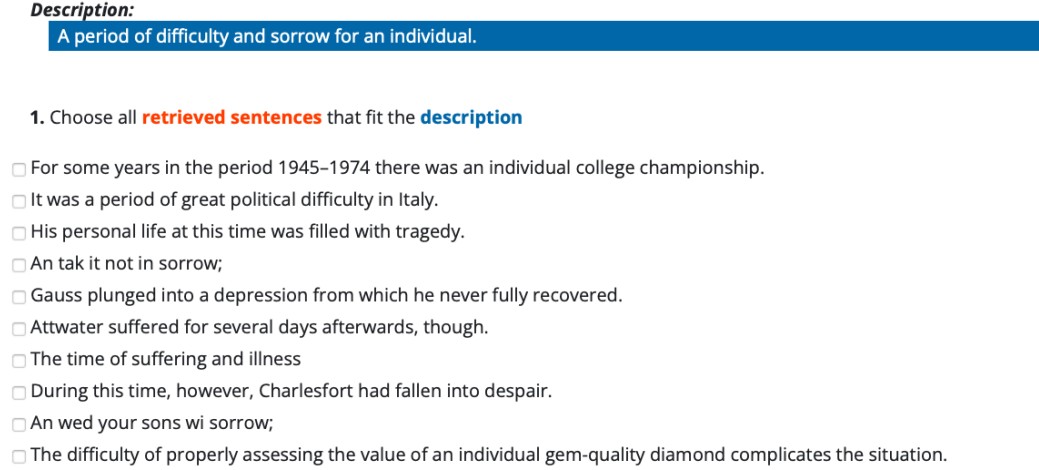

This is the interface we used for assessing the coverage of the GPT3 generated description and its corresponding sentence.

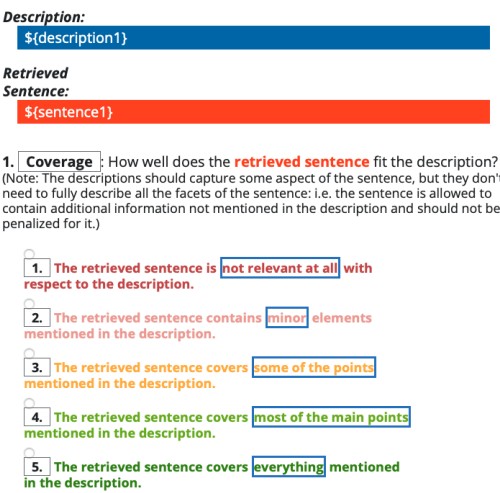

