# OpenReview forum: "Description-Based Text Similarity"
_colmweb.org/COLM/2024/Conference — COLM_

### Official Review · Reviewer_i7FB · 2024-05-06

**Rating:** 7
**Confidence:** 3
**Ethics Flag:** 1

**Summary:**

This paper argues that current embedding similarity approaches are inadequate for retrieval. Sentence embeddings do not necessarily abstract over the meaning of a text and hence during retrieval we cannot simply use similarity metrics over the actual embeddings of the text. We instead need to develop some summary / description / abstraction of this text and use this to guide our search. The authors construct a data set to test this hypothesis.

**Reasons To Accept:**

I think that the paper merits acceptance. The hypothesis of the paper is intuitive, and there is enough novelty (including the construction of a data set) to support publication.

**Reasons To Reject:**

The main flaw of the paper is in the assessment of the method. The data set is not large enough to assess how their approach could assist in more accurate document retrieval in more practical terms. Nevertheless, I understand that constructing such a data set is not an easy endeavour.

---

> ### Author Rebuttal · Authors · 2024-05-29
>
> Thank you for the positive feedback and recognition of the novelty of our work.
>
> ***Dataset Size and Practical Implications***: We acknowledge that the current dataset may not be large enough to fully assess the practical implications of our approach in broader document retrieval scenarios, however, we put emphasis on the quality of the queries used in the evaluation, as we manually verified each of them, and edited them when it was necessary. While the dataset, consisting of 200 queries, is not very long, it is still long enough to showcase the limitations of existing approaches to the problem.

---

> > ### Comment · Reviewer_i7FB · 2024-06-01
> >
> > The authors have addressed my concerns. That does not affect my score.

---

> > > ### Author Response · Authors · 2024-06-05
> > > **Response**
> > >
> > > Thank you!

---

### Official Review · Reviewer_9ziQ · 2024-05-09

**Rating:** 4
**Confidence:** 3
**Ethics Flag:** 1

**Summary:**

This paper presents a new variant of information retrieval where the query is an abstract description and the goal is to return specific instances that match the description. For example, if the description is "A neurotransmitter found in the brain in high concentrations." then an acceptable result will be the sentence "Dopamine constitutes about 80% of the catecholamine content in the brain.".

The method in the paper is to train special sentence encoders using a special dataset of triples of acceptable result, good description and bad description. LLMs (GPT-3) are used to generate the descriptions and a dual-encoder method to embed the abstract description which is the query, and the document using two separate models. The loss function was empirically determined to be a combination of triplet loss and info-nce loss.

**Reasons To Accept:**

This paper presents a variant of the fairly standard task of information retrieval given a long query. The paper is well written and readable and the results are plausible. This work critically relies on LLMs for bootstrapping its dataset and it might be helpful to practitioners working on this problem.

**Reasons To Reject:**

The paper lacks technical depth and the methods and results are not very insightful. The task presented in this paper is claimed to be novel but it is quite similar to the long query retrieval tasks at TREC. Also the subject of the paper is only tangentially related to language modeling. The main use of LLMs in this paper is to generate the training dataset.

Overall the proposed method does not seem to be very novel or technically significant and has only a little incremental value. Therefore I will not recommend to accept this paper for publication at COLM.

---

> ### Author Rebuttal · Authors · 2024-05-29
>
> Thank you for the feedback and the opportunity to address the concerns raised. Our task of retrieving text based on abstract descriptions introduces a new notion of similarity. It has no relation to TREC's long query tasks.
>
> Unlike TREC's long query tasks, which generally focus on detailed and verbose queries directly related to the content, our work addresses the challenge of aligning abstract, high-level descriptions with specific instances. This requires a different approach and demonstrates significant novelty in the context of information retrieval. We do not understand in what sense this is “incremental”.

---

> > ### Comment · Reviewer_9ziQ · 2024-06-04
> > **Rejoinder to Rebuttal**
> >
> > I am clearly in the minority regarding whether the paper should be accepted or not, but I still don't understand the technical merit of the paper and its relevance to the conference. The only usage of an LM is to generate the training dataset, and the methods used such as triplet loss, and dual encoder methods are fairly standard practice in industry now. I am unfortunately still missing the significance of this work and can not raise the score.

---

> > > ### Author Response · Authors · 2024-06-05
> > > **Response**
> > >
> > > Thanks for responding. To help us understand the issue, could you please clarify whether our response explains the novelty in the *task* itself, which is different than previously proposed retrieval tasks? We tried to explain why retrieval by abstract descriptions is (1) new, (2) important, and (3) not done well by current SOTA models *withour* data tailored for this task by LMs.

---

### Official Review · Reviewer_Utoh · 2024-05-10

**Rating:** 7
**Confidence:** 4
**Ethics Flag:** 1

**Summary:**

This paper identifies the need to search for texts based on abstract descriptions of their content, and the corresponding notion of description-based similarity. An embedding model combining the triplet loss and the InfoNCE loss is proposed, and it can significantly improve the standard nearest neighbor search.

**Questions To Authors:**

•	To my knowledge, InfoNCE usually sets $\tau$ to 0.05, but this paper sets it to 0.1. I haven't seen an ablation study to support this setting. Could you explain this further?
•	Why do you use GPT-3 instead of more powerful LLMs like ChatGPT to generate datasets?

In addition, I put some suggestions and typo correction comments as follows.

Suggestions
•	It is better to provide an explanation for the hyperparameter selection.
•	It is better to present the statistics, such as train/dev/test set size, positive and negative size, of the proposed data in a table.
•	There might be some missing references in the section on obtaining training data. LLM-based synthetic data for text similarity or information retrieval have been explored [1][2]. It is better to also acknowledge their contributions.
[1] Li X, Li J. Angle-optimized text embeddings[J]. arXiv preprint arXiv:2309.12871, 2023.
[2] Wang L, Yang N, Huang X, et al. Improving text embeddings with large language models[J]. arXiv preprint arXiv:2401.00368, 2023.

Typos:
Page 3: iff -> if (in definition 1)
Page 3: londer -> longer (in footnote 1)
Page 14: ?? missing reference?

**Reasons To Accept:**

•	This paper presents a new type of semantic textual similarity task: description-based text similarity. This new type of similarity benefits retrieval-augmented generation.
•	This paper proposes a description-based text similarity dataset. This dataset is helpful for this research area.
•	Both the human and auto evaluations suggest the effectiveness of the proposed model.

**Reasons To Reject:**

•	The experiment is inadequate. The experiments focus on comparing different models' performance. However, there is no experiment to support the significance of the proposed new type of similarity task. It would be appreciated if you could provide some experiments on downstream tasks to quantitatively demonstrate the importance of the proposed description-based text similarity task.
•	The experiment might be unfair. It is better to also compare the proposed model with baselines fine-tuned on the collected dataset.

---

> ### Author Rebuttal · Authors · 2024-05-29
>
> Thank you for the insightful feedback.
>
> ***Significance of Description-Based Similarity Task***: We envision the task of retrieving texts by their description to be directly useful on its own for many end-uses, in particular information seeking by experts in large document collections. This is true regardless of any relevance for an existing NLP end-task.  A second-order user-study can be useful, but is not in the scope of the current work.
>
> In the introduction section, we discuss the use cases of our proposed approach in information-seeking scenarios. Its Usefulness is particularly highlighted in scenarios where experts aim to locate domain-specific relevant text in large corpora. For instance, a legal researcher can inquire about "precedents for intellectual property disputes". Natural language descriptions provide an effective way for domain experts to search over large corpora of natural language texts.
>
> ***Fair Comparison with Fine-Tuned Baselines***: We focus on drawing attention to the need to retrieve by description, and showcasing the limitations of existing encoders in this task. We find it essential to test models on our data to showcase the limitations of SOTA general-purpose encoders in capturing the notion of similarity we argue is useful. Their failure is a nontrivial finding, as these models were trained on orders of magnitude more data. It is not possible to highlight this inherent limitation of existing models without evaluating them on data tailored for our notion of similarity.
>
> ***InfoNCE Temperature Setting***: The choice of the temperature parameter in InfoNCE was based on preliminary experiments where a value of 0.1 yielded better performance for our specific task. We acknowledge the need for an ablation study to support this choice and will include such a study in the revised manuscript to provide a detailed justification.
>
> ***Choice of GPT-3 over ChatGPT***:  Please note that we used text-davinci-003, an instruction-tuned version. We found that it worked better than ChatGPT, which generated highly verbose responses. However, the LM choice is pretty much orthogonal to our approach.
>
> ***Dataset Statistics***: We will present detailed statistics of our dataset, including train/dev/test set sizes and the number of positive and negative examples, in a table for better clarity.
>
> ***Missing References***: We appreciate the pointers to relevant literature. We will include the suggested references on LLM-based synthetic data generation.

---

> > ### Comment · Reviewer_Utoh · 2024-06-04
> > **Response to authors' rebuttal**
> >
> > Dear authors,
> > ﻿
> > Thank you for your explanation; my previous concerns have been addressed. As a result, I decide to raise the score to 7.
> > I believe the newly proposed task and dataset will be helpful for the text similarity research. Hopefully, you will include the missing experiments and references in the next version.
> >
> > Best,
> > Reviewer Utoh

---

> > > ### Author Response · Authors · 2024-06-05
> > > **Response**
> > >
> > > Thank you!

---

### Official Review · Reviewer_88m7 · 2024-05-10

**Rating:** 6
**Confidence:** 4
**Ethics Flag:** 1

**Summary:**

The paper tackles the question of what would be a good similarity measure for effective text retrieval, suggesting that the similarity induced from word embeddings is corpus driven (trained on sentence pairs labeled as similar according to some notion of similarity, which is not always consistent) and not optimal for many use-cases.

The authors propose a new model that works better then current embeddings in nearest neighbor search. Such a model is trained using LLMs to annotate/describe pieces of text and generate positive and negative examples.

The key idea is to obtain an abstract description of a sentence, where the sentence participates in an instance-of relation with its description.

Current dense retrieval methods tend to retrieve texts that are similar to the description, so a model that is specialized on retrieving instantiation of the description is needed.

The authors train an encoder model that maps <description, text> pairs close in the embedding space. The description of a piece of text is obtained with the davinci-text-03 model. The model produces different descriptions, and some that are incorrect are used as a negative example.

**Questions To Authors:**

Encoders for descriptions and sentences have separate weights because of the asymmetry of the task. Have you tried tying the weights?

Suggestions: in Figure 2, the colors used for the Abstract-sim and Instructor bars are too similar and difficult to separate for some people. Please use a different color scheme.

Typos: soltuion, desciprion

**Reasons To Accept:**

The idea of using LLMs to create special kinds of queries related to a piece of content and then train a specialized retrieval model on such data is interesting and it has a lot of potential.

**Reasons To Reject:**

I think the ways to evaluate the proposed approach are tricky and the current system definitely has some shortcomings, one of them being too narrow and favorable to the proposed fine-tuned model.

The authors are defining a task that is focused on a subset of all possible search queries, they are extracting features (descriptions) of sentences via an automatic process (LLM), training a model on such data and testing using descriptions that are automatically generated too.

Therefore, the model is by definition specialized for this task (and the other baselines are not) and in addition, test descriptions are in-distribution. Probably a manually designed set would have been a little better.

Maybe expanding the type of queries and showing an approach that can improve over some types of queries without hurting the performance of the others would be more interesting.

---

> ### Author Rebuttal · Authors · 2024-05-29
>
> Thank you for the detailed and thoughtful feedback.
>
> ***Specialization and Fairness***: While our model is indeed specialized for the task of retrieving text by abstract description, this specialization underscores a critical point: existing models, trained on broader and often less specific definitions of similarity (I.e., many different similarity datasets), fail to capture this notion, as demonstrated in our experimental evaluation. The in-distribution test data was necessary to fairly evaluate the capability of our model in this specific context. Out main contribution is not the training of the specific model (indeed, it is very likely that better models can be found), but rather in advocating for this specific notion of similarity and showing that current models fall short in acquiring it. Please note that the test set was sourced from GPT-generated queries, but was further manually evaluated and refined by the authors.
>
> ***Tying encoders weights***: We have tried that in preliminary experiments, but this resulted in somewhat decreased performance. Please note that in our setting, the queries and the results are very different in their surface form: queries are short and highly abstract, while results are concrete and detailed
>
> ***Color Scheme in Figure 2***: We apologize for the oversight in our color selection. We will revise Figure 2 to use a more distinct and accessible color scheme.

---

> > ### Comment · Reviewer_88m7 · 2024-06-05
> >
> > Thank you for addressing my comments. I think the current score reflects my level of excitement about the paper and the evaluation, which at the essence shows that a dual encoder model can retrieve data with some distinctive properties if fine-tuned on it. I agree that this notion of similarity is interesting and orthogonal to others. I would have really liked to see the usefulness of the generated data being explored in a different setting.

---

> > > ### Author Response · Authors · 2024-06-05
> > > **Response**
> > >
> > > Thanks for the explanation. We agree that intrinsic evaluation of the contribution of retrieval by description is an important future work.

---

### Decision · Program_Chairs · 2024-07-10

**Decision:**

Accept

**Comment:**

The reviewers generally like the idea proposed in this paper, which leverages LLMs to create new data for training a specialized retrieval model. Most of the criticisms center on the evaluation, including the size of the dataset, the impact on downstream tasks, etc. The authors should address these feedback in the next version of this paper properly.